# Mechanochemical synthesis of pillar[5]quinone derived multi-microporous organic polymers for radioactive organic iodide capture and storage

Kecheng Jie[1,2], Yujuan Zhou[3], Qi Sun[4], Bo Li[5], Run Zhao[3], De-en Jiang [5], Wei Guo[1,2], Hao Chen[1,4], Zhenzhen Yang[1,2], Feihe Huang[3] & Sheng Dai [1,2✉]

The incorporation of supramolecular macrocycles into porous organic polymers may endow the material with enhanced uptake of specific guests through host−guest interactions. Here we report a solvent and catalyst-free mechanochemical synthesis of pillar[5]quinone (P5Q) derived multi-microporous organic polymers with hydrophenazine linkages (MHP-P5Q), which show a unique 3-step $N_2$ adsorption isotherm. In comparison with analogous microporous hydrophenazine-linked organic polymers (MHPs) obtained using simple twofold benzoquinones, MHP-P5Q is demonstrated to have a superior performance in radioactive iodomethane ($CH_3I$) capture and storage. Mechanistic studies show that the rigid pillar[5]arene cavity has additional binding sites though host−guest interactions as well as the halogen bond ($-I\cdots N=C-$) and chemical adsorption in the multi-microporous MHP-P5Q mainly account for the rapid and high-capacity adsorption and long-term storage of $CH_3I$.

[1] Department of Chemistry, The University of Tennessee, Knoxville, TN 37996-1600, USA. [2] Chemical Sciences Division, Oak Ridge National Laboratory, Oak Ridge, TN 37831-6201, USA. [3] State Key Laboratory of Chemical Engineering, Center for Chemistry of High-Performance & Novel Materials, Department of Chemistry, Zhejiang University, 310027 Hangzhou, P. R. China. [4] College of Chemical and Biological Engineering, Zhejiang University, 310027 Hangzhou, P. R. China. [5] Department of Chemistry, University of California, Riverside, CA 92521, USA. ✉email: dais@ornl.gov

Supramolecular macrocycles including crown ethers, cyclodextrins, calixarenes, cucurbiturils, and pillararenes have been known as a kind of supramolecular receptors with intrinsic cavities to bind specific guest molecules in solution[1–6]. Other than the solution-based host–guest chemistry, some supramolecular macrocycles such as calixarenes[7] and pillararenes[8] exhibit intrinsic microporosity in the solid state for guest adsorption owing to their pre-fabricated cavities of a certain type. Particular attention should be paid to the recent advances in pillararene-based nonporous adaptive crystals (NACs)[9–13]. Specific guests will trigger the structural change of these nonporous crystals, thus recovering the intrinsic cavities of pillararenes to accommodate guests. This unique property makes NACs of pillararenes promising adsorptive materials in highly selective separations and environment treatment[9–13]. However, the practical applications of pillararene NACs in these aspects suffer from some inherent shortcomings, such as the slow adsorption rate due to the slow guest-induced structural change process and the limited adsorption amount in the intrinsic cavity.

Porous organic polymers (POPs) are a class of porous materials featuring high surface areas, low densities, synthetic diversity, high chemical/thermal stabilities, etc.[14–17]. In light of these unique merits, a variety of POPs including conjugated microporous polymers (CMPs)[18–20], polymers of intrinsic microporosity[21,22], porous aromatic frameworks[23,24], microporous hypercrosslinked polymers[25,26], covalent triazine frameworks (CTFs)[27,28], and covalent organic frameworks (COFs)[29–31] have been synthesized to explore their full applications in molecular separations, gas storage, catalysis, environmental treatment, and so on. Recently, supramolecular macrocycles have been incorporated into POPs to realize specific applications[32–34]. However, these macrocycles usually act as multi-fold crosslinkers while their pre-fabricated cavities have rarely been demonstrated in POPs presumably due to their flexible skeletons (the cavity is not permanent) or monomer penetration into the cavity. We proposed that the incorporation of pillararene backbones with the reservation of their cavities into POPs might overcome the shortcomings of NACs and combines the properties and advantages of NACs and POPs. Specifically, such material might generate multiple binding sites (intrinsic pillararene cavity for host–guest interactions and extrinsic functional binding sites) for the maximization of specific guest-loading as well as abundant multi-micropores (intrinsic and extrinsic space) for rapid guest transportation by avoiding slow structural transitions.

To this end, we aimed to prepare a POP derived from pillar[5] quinone (P5Q) with a pre-fabricated cavity, through a facile aza-ring formation reaction between ortho-diamines and the repeating benzoquinone units of P5Q[35]. However, the poor solubility of P5Q in common organic solvents greatly restricts the solution-based synthesis of P5Q-derived POPs. Particular attention was thus paid to mechanochemistry, which has been demonstrated to be a powerful tool in the synthesis of porous materials including metal-organic frameworks, zeolitic imidazolate frameworks and POPs as well[36–41]. Compared with solution-based synthetic procedures in the presence of high-performance catalysts, the solvent-free mechanochemical approach shows superiority in rapid, scalable, and environmentally friendly preparation regardless of the solubility of the monomers. Herein, we report a solvent and catalyst-free mechanochemical synthesis of P5Q-derived multi-microporous organic polymers with hydrophenazine linkages (MHP-P5Q), which displays a unique three-step $N_2$ sorption isotherm with three distinct pore size distribution. In specific, triptycenehexamine (THA) with a unique internal free volume feature is employed as a threefold crosslinker to react with P5Q, a 10-fold monomer, to afford MHP-P5Q. In contrast to analogous microporous hydrophenazine-linked organic polymers (MHPs)

obtained using simple benzoquinones (twofold monomers) and THA, MHP-P5Q with similar Brunauer–Emmett–Teller (BET) surface area has a superior performance in the capture and storage of $CH_3I$, a typical radioactive organic iodide waste produced in the nuclear industry. Mechanistic studies confirm that the rigid pillar [5]arene cavity as additional binding sites through host–guest interactions as well as the halogen bond (–I···N=C–) and chemical adsorption in the multi-microporous MHP-P5Q play a vital role in the rapid and high-capacity adsorption and long-term storage of $CH_3I$.

## Results

**Mechanochemical synthesis of model compound and MHPs.** We firstly investigated the possibility to synthesize an aza-fused hydrophenazine-like model compound (MC) using *o*-phenylenediamine (*o*-PDA) and benzoquinone (BQ) via mechanochemistry. Ball milling of *o*-PDA with BQ for 30 min afforded rigid MC with a yield of over 95% (Supplementary Figs. 4–7), which may be effective for the mechanochemical synthesis of MHPs. Afterward, hydrophenazine-linked microporous organic polymers MHP, MHP-Cl, and MHP-Br were prepared by ball milling of THA (threefold monomers) with simple benzoquinones (twofold monomers) including BQ, 2,5-dichloro-1,4-benzoquinone (BQ-Cl), and 2,5-dibromo-1,4-benzoquinone (BQ-Br) for 30 min, respectively (Fig. 1a). To the best of our knowledge, such a hydrophenazine formation reaction has never been realized via a mechanochemical approach[34], let alone in the mechanochemical synthesis of porous polymers. Based on the above, P5Q-derived hydrophenazine-linked microporous organic polymer (MHP-P5Q) was synthesized via the same mechanochemical approach by the substitution of twofold BQs with 10-fold P5Q (Fig. 1b).

**Physiochemical characterization of MHPs.** The structures of these insoluble polymeric solids were firstly characterized by solid-state $^{13}C$ cross-polarization magic-angle spinning nuclear magnetic resonance ($^{13}C$ CP-MAS NMR). The $^{13}C$ CP-MAS NMR spectrum of MHP-P5Q reveals nine carbon peaks with chemical shifts of 201.9, 171.9, 140.7, 129.4, 121.9, 111.9, 81.4, 62.5, and 53.1 ppm, which are assigned to the edge carbonyl (C=O) groups (a), the aromatic ($sp^2$) carbons (b, c, d, e, f, g), and the $sp^3$ bridge carbon (h, i), respectively (Fig. 2a). It is worth noting that the other three samples have identical spectra, which are also similar to that of MHP-P5Q (Supplementary Figs. 8, 11 and 14). An exception for MHP-P5Q is the peak at 62.5 ppm (methylene-bridged carbon on P5Q), which does not exist in the spectra of the other three samples (Fig. 2a). Elemental analysis reveals a nitrogen content of 15.93, 16.25, 13.18, and 10.44 wt% for MHP-P5Q, MHP, MHP-Cl, and MHP-Br, respectively, indicating the existence of abundant nitric groups in MHPs. X-ray photoelectron spectroscopy (XPS) was performed to probe the nitrogen bonding nature in MHPs. The spectrum of MHP-P5Q shows two N 1s peaks at 398.88 and 400.48 eV, which can be attributed to the characteristic imine nitrogen atoms and aniline nitrogen atoms[42], respectively (Fig. 2b). Meanwhile, the two peak areas are almost the same, indicating the identical content of the two nitrogen species in MHP-P5Q. Similar results are also observed for MHP, MHP-Cl, and MHP-Br (Supplementary Fig. 18), revealing the formation of hydrophenazine rings in MHPs. Fourier-transform infrared spectroscopy (FT-IR) showed that MHP-P5Q and MHP have several similar peaks at 1713/1712, 1624/1614, and 1356/1351 cm$^{-1}$, corresponding to the stretching vibrations of C=O, C=N, and C–N, respectively (Fig. 2c). Such characteristic peaks can also be found in the spectra of MHP-Cl and MHP-Br (Supplementary Fig. 19). These results are thus consistent with XPS and $^{13}C$ CP-MAS NMR results. The thermal stability of MHPs

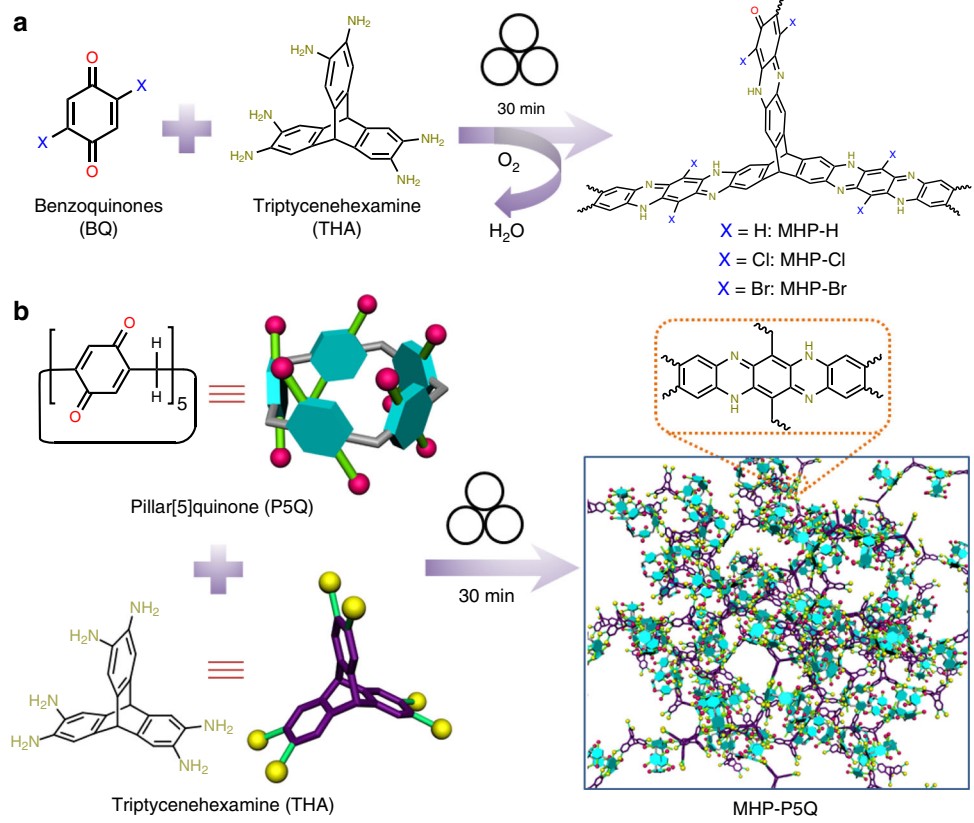

**Fig. 1 The synthesis of hydrophenazine-linked porous polymers via mechanochemistry. a** Chemical structures and synthetic routes to MHP, MHP-Cl, and MHP-Br. **b** Schematic representation of the synthesis of MHP-P5Q using P5Q and THA.

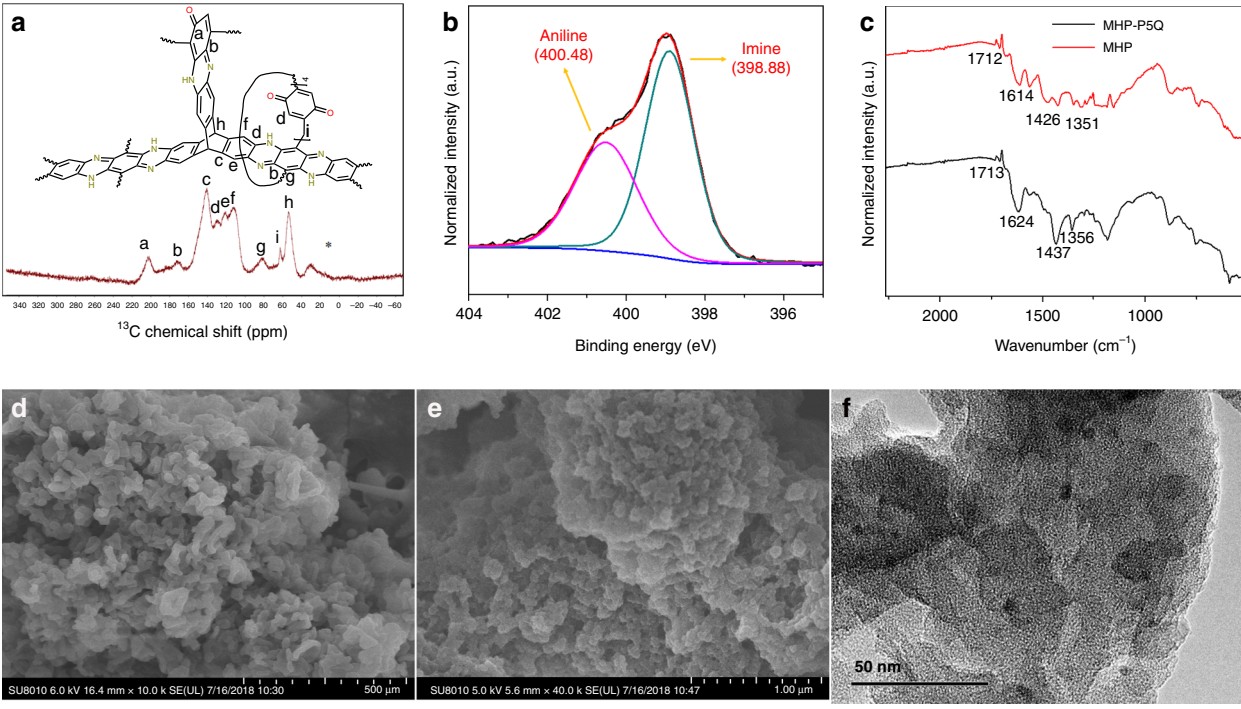

**Fig. 2 Structural characterizations of MHP-P5Q. a** Solid-state $^{13}$C CP-MAS NMR spectrum. The asterisk denotes spinning sidebands. **b** N 1s XPS spectrum. **c** FT-IR spectra of MHP and MHP-P5Q. **d, e** FE-SEM images at different scales. **f** HR-TEM image.

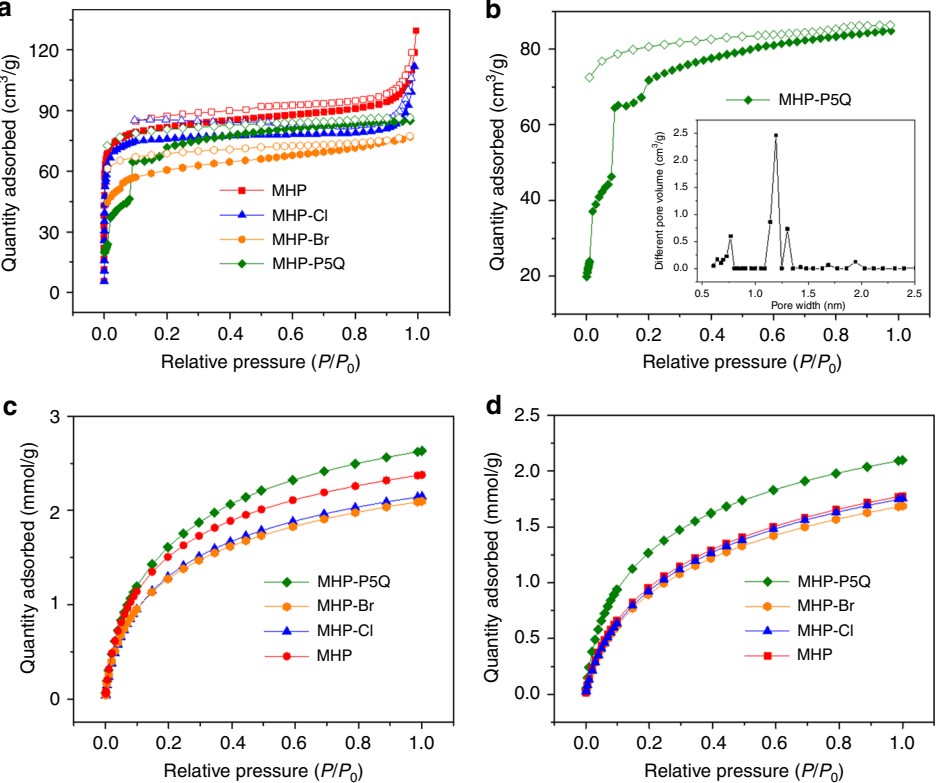

**Fig. 3 Gas sorption properties of MHPs. a** $N_2$ adsorption and desorption isotherms of MHP (red, $S_{BET} = 320$ m²/g, MHP-Cl (blue, $S_{BET} = 289$ m²/g), MHP-Br (orange, $S_{BET} = 208$ m²/g), and MHP-P5Q (green, $S_{BET} = 296$ m²/g); **b** $N_2$ adsorption and desorption isotherms of MHP-P5Q at 77 K (inserted: corresponding pore size distribution from the NLDFT approximation). Solid symbol: adsorption; open symbol: desorption. $S_{BET}$ denotes BET surface areas. $CO_2$ adsorption isotherms of MHP, MHP-Cl, MHP-Br, and MHP-P5Q at **c** 273 K and **d** 298 K.

was determined by thermogravimetric analysis (TGA). All four MHPs were demonstrated to be stable below around 330 °C, indicating their good thermal stability (Supplementary Figs. 10, 13, 16, and 21).

Powder X-ray diffraction (PXRD) experiments were performed to investigate the long-range ordering of MHPs. No sharp peaks were observed in the PXRD spectra for all MHPs, revealing their amorphous nature (Supplementary Figs. 9, 12, 15, and 20). Meanwhile, the bulk morphologies of MHP-P5Q were visualized with field-emission scanning electron microscopy (FE-SEM). The SEM images show that MHP-P5Q consists of relatively uniform solid sub-micron spheres (Fig. 2d, e). These fused polymer masses without well-defined shapes also imply the loss of long-range order. The internal structure of MHP-P5Q was visualized by high-resolution transmission electron microscopy (HR-TEM). No ordered structures or crystal lattice can be observed at the nanoscale, indicating the loss of crystallinity (Fig. 2f).

**Gas sorption studies.** Nitrogen sorption experiments at 77 K were then carried out to evaluate the porosity of MHPs. As shown in Fig. 3a, MHP, MHP-Cl, and MHP-Br display typical type I sorption isotherms, indicating the microporosity of the three materials. Intriguingly, a unique adsorption isotherm with three distinct steps can be observed for MHP-P5Q (Fig. 3b), which is often characteristic of porous materials with a uniform distribution of different pore sizes[43]. As far as we know, this phenomenon has barely been observed in amorphous porous polymers. The BET surface areas of MHP, MHP-Cl, MHP-Br, and MHP-P5Q are calculated to be 320, 289, 208, and 296 m²/g, respectively,

whereas their calculated total pore volumes are 0.16, 0.14, 0.11, and 0.13 cm³/g, respectively. The pore size distributions of MHP, MHP-Cl, and MHP-Br determined by NLDFT all showed multiple peaks ranging from 0.7 to 2.0 nm, indicating the nonuniform pore size distribution (Supplementary Figs. 22−24). In contrast, the pore size distribution of MHP-P5Q showed three distinct peaks with pore diameters at 0.76, 1.19, and 1.30 nm, respectively, implying the uniform multi-microporosity (Fig. 3b). It is worth noting that the pore diameter at 0.76 nm is consistent with the cavity size of pillar[5]arene[44], which on the other hand confirms the presence of the permanent pillar[5]arene cavity in the robust frameworks. Meanwhile, all the MHPs show abundant microporosity as the ratio of micropore volume to the total pore volume ($V_{micro}/V_{total}$) is assessed to be 0.69, 0.75, 0.73, and 0.77 for MHP, MHP-Cl, MHP-Br, and MHP-P5Q, respectively (Supplementary Table 2).

To better understand the formation mechanism of the multi-microporous structure of MHP-P5Q, samples of P5Q-TAB and P5Q-DAB were obtained using twofold diamines including 1,2,4,5-tetraaminobenzene (TAB) and 3,3′-diaminobenzidine (DAB) with P5Q via mechanochemistry. However, the two samples were demonstrated to be barely porous (Supplementary Fig. 25), indicating that the threefold THA also played a vital role in the construction of multi-microporous polymer MHP-P5Q. On the one hand, the reaction between P5Q and threefold THA endows MHP-P5Q with 3D framework structures, thus generating extrinsic microporosity. On the other hand, THA with a huge fragment can avoid monomer penetration into P5Q cavity and inhibit the rotation of the P5Q repeating units as well, thus leading to the preservation of intrinsic P5Q cavities in the 3D framework. In contrast, the reactions of linear twofold diamines

with P5Q are likely to form linear polymers, resulting in the loss of porosity.

After evaluation of the porosity, the $CO_2$ adsorption capabilities of MHPs were also investigated. The $CO_2$ uptake amount at 1 bar and 273 K for MHP, MHP-Cl, MHP-Br, and MHP-P5Q are 2.37, 2.14, 2.09, and 2.63 mmol/g, respectively, while those at 1 bar and 298 K are 1.77, 1.75, 1.68, and 2.09 mmol/g, respectively (Fig. 3c, d). The results reveal that although the BET surface area of MHP-P5Q is lower than MHP, it has the best $CO_2$ capture performance among the four MHPs. This may be ascribed to the synergistic effects of the multi-microporosity in MHP-P5Q for $CO_2$ transportation and the abundant $CO_2$-philic surface sites, including unreacted C=O sites and nitric sites[45].

**Radioactive organic iodide capture performance.** Nuclear energy has been growing rapidly to meet the increasing demand for global energy. One challenge to be faced with is the capture and reliable storage of nuclear waste including radioactive molecular iodine ($I_2$) and organic iodides (e.g., iodomethane and iodoethane) in the production to ensure safe nuclear energy usage[46,47]. The capture of radioactive organic iodides is particularly challenging due to their high volatility[46–50]. We envisioned that MHP-P5Q with abundant nitric groups and multi-micropore structures might be an ideal adsorbent to capture and store radioactive organic iodides. Other MHPs as well as perethylated pillar[5]arene (EtP5) crystals (Supplementary Fig. 35), a kind of well-investigated NACs, were also investigated as contrasts.

To explore the adsorption capacities in these adsorbents, iodomethane ($CH_3I$) vapor sorption isotherms of MHPs were firstly obtained. As can be seen from Fig. 4a, the $CH_3I$ vapor

adsorption amount in MHP-P5Q is the highest (218 $cm^3/g$) at 1 bar and 298 K, while those in MHP, MHP-Cl, MHP-Br, and EtP5 are significantly lower with the values of 143, 110, 102.5, and 55 $cm^3/g$, respectively. In specific, the uptake of $CH_3I$ in EtP5 shows a typical gate-opening behavior with the gate-opening pressure at $P/P_0 = 0.7$ (Fig. 4a, purple squares), which has been observed previously for pillararene NACs in the adsorption of other hydrocarbons[9–13]. It is notable that distinct hysteresis in the desorption process can be observed for all five adsorbents, indicating the strong interactions between $CH_3I$ and these adsorbents. Meanwhile, a certain amount of the adsorbed $CH_3I$ is retained in the adsorbents even under reduced pressure (Fig. 4a). For MHP-P5Q, the reserved amount is 118 $cm^3/g$, which is about 54% of the absorbed amount (Fig. 4a, b). The reserved amounts for MHP, MHP-Cl, and MHP-Br are 67, 51, and 48 $cm^3/g$, respectively, which are less than half of the adsorbed amount (Fig. 4a, b). For EtP5, the reserved amount is quite small (9 $cm^3/g$), indicating the less stable storage of $CH_3I$ (Fig. 4a, b). The above results imply that MHP-P5Q has the highest $CH_3I$ capture capacity and the potential in the storage of the captured $CH_3I$.

To demonstrate the ability of these adsorbents in the capture of $CH_3I$ from the air, time-dependent solid–vapor adsorption experiments were then carried out. Similar to the vapor sorption isotherm results, MHP-P5Q was demonstrated to have the highest uptake amount of $CH_3I$ (80.3 wt%), followed by MHP (62.2 wt%), MHP-Cl (52.8 wt%), MHP-Br (49.2 wt%), and EtP5 (27.6 wt%; Fig. 4c). Meanwhile, the uptake rate of $CH_3I$ in MHP-P5Q is also the highest among these adsorbents. It took only 1 h for MHP-P5Q to reach the saturation point while the time for other MHPs was 2 h (Fig. 4c). A longer time (6 h) was needed for EtP5 to reach the saturation point, a typical phenomenon for

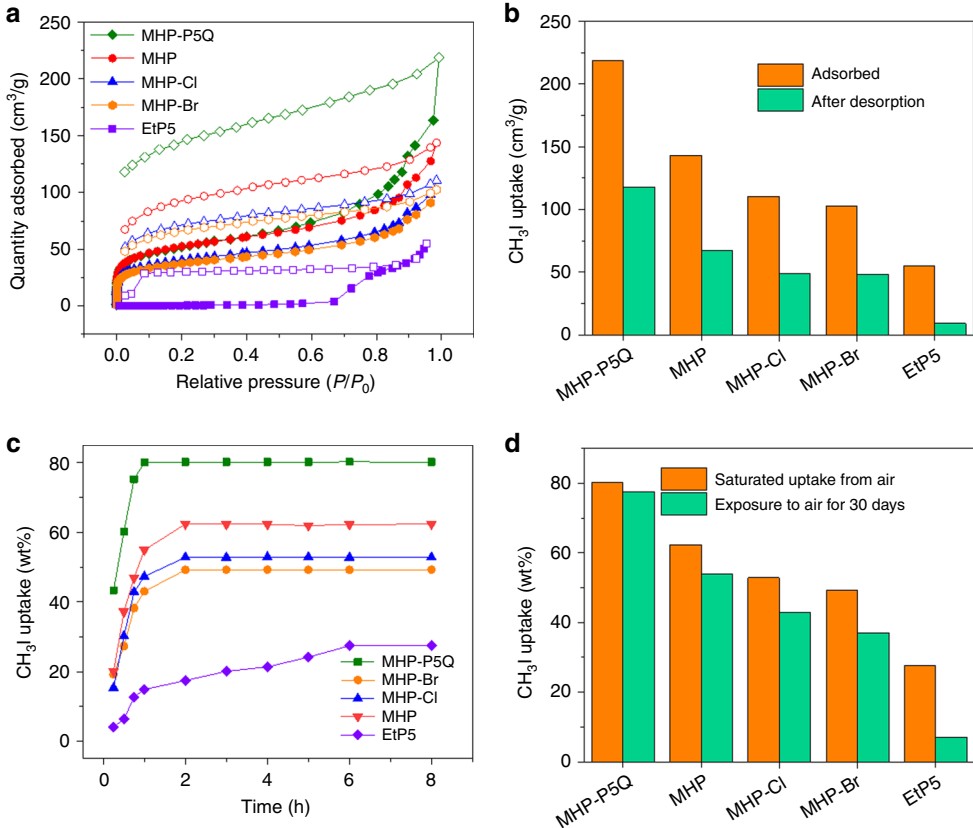

**Fig. 4 $CH_3I$ capture experiments. a** $CH_3I$ vapor adsorption and desorption isotherms of EtP5 (purple), MHP (red), MHP-Cl (blue), MHP-Br (orange), and MHP-P5Q (green). Solid symbol: adsorption; open symbol: desorption. **b** Uptake amount of $CH_3I$ at 1 bar and 25 °C and reserved amount after desorption. **c** Time-dependent adsorption amount of $CH_3I$ vapor in EtP5 (purple), MHP (red), MHP-Cl (blue), MHP-Br (orange), and MHP-P5Q (green) at 25 °C. **d** Saturated $CH_3I$ uptake from the air and after exposure to air for 30 days at 25 °C.

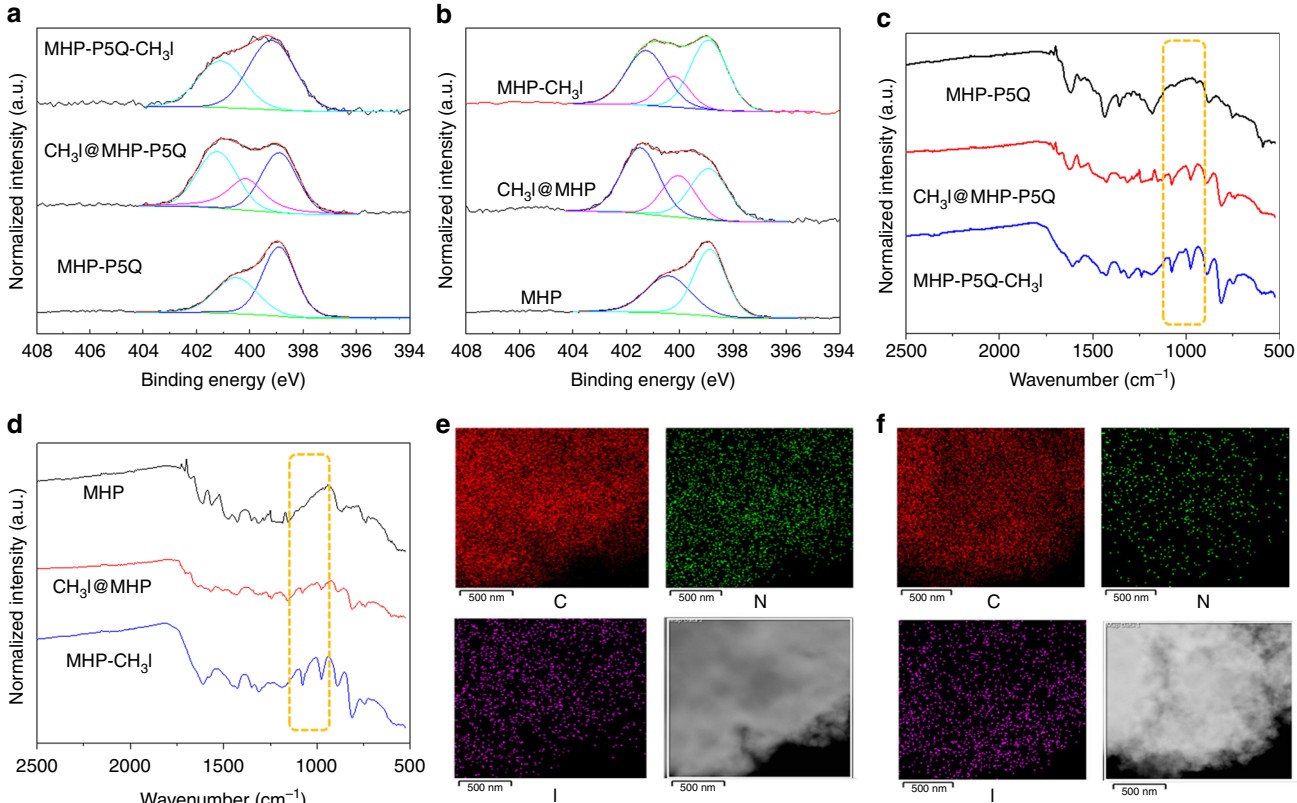

**Fig. 5 Mechanistic studies on the CH₃I capture.** N 1*s* XPS spectra of **a** MHP-P5Q, CH₃I@MHP-P5Q, and MHP-P5Q-CH₃I, and **b** MHP, CH₃I@MHP, and MHP-CH₃I. FT-IR spectra of **c** MHP-P5Q, CH₃I@MHP-P5Q, and MHP-P5Q-CH₃I, and **d** MHP, CH₃I@MHP, and MHP-CH₃I. The elemental mapping of **e** CH₃I@MHP-P5Q and **f** CH₃I@MHP after heating at 100 °C for 2 h.

these nonporous crystals due to the slow structural transformations (Fig. 4c). We also compared the CH₃I adsorption capacities of MHPs with several benchmark adsorbents that are actually used in nuclear industry, such as activated porous carbon impregnated with triethylenediamine (TED@AC) and silver functionalized zeolites (including ZSM-5, 13×, and mordenite, which are named as Ag⁺@ZSM-5, Ag⁺@13×, Ag⁺@MOR, and Ag0@MOR for short, respectively, Supplementary Fig. 27). The results confirm that MHP-P5Q has the best performance in CH₃I capture from air, which may be beneficial for its potential use in nuclear energy production.

Another practical task for adsorbent is the reliable storage of the radioactive organic iodides after they are captured from nuclear wastes. Leakage of these radioactive species from the adsorbents may severely damage not only the staff in the nuclear energy industry but also the surrounding residents and environment[40]. In order to demonstrate the long-term storage of CH₃I in these adsorbents, the CH₃I-loaded samples were exposed to the atmosphere for 30 days at room temperature. Gravimetric analyses showed that the reserved amount of CH₃I in MHP-P5Q was 77.6 wt%, meaning that more than 97% of the adsorbed CH₃I was stably stored in MHP-P5Q (Fig. 4d). In contrast, the reserved CH₃I percentage in MHP, MHP-Cl, and MHP-Br is 86.7%, 81.4%, and 75.2%, respectively, while that in EtP5 is only 25.4% (Fig. 4d). The above experimental results suggest that the captured CH₃I can be stored in MHP-P5Q with exceptional long-term stability, which remarkably exceeds other adsorbents.

## Discussion

The exceptional CH₃I capture performance and particularly stable CH₃I storage in MHP-P5Q prompted us to investigate the possible mechanism. Hence, the structures of CH₃I-loaded MHP-P5Q (CH₃I@MHP-P5Q) as well as CH₃I-loaded MHP (CH₃I@MHP) were characterized by solid-state TGA, XPS, and FT-IR experiments. Surprisingly, N 1*s* XPS spectra of both CH₃I@MHP-P5Q and CH₃I@MHP show three distinct peaks, which are distinctly different from pristine MHP-P5Q and MHP (Fig. 5a, b). The peaks corresponding to the nitrogen atoms on the imine group remain the same while the peaks related to the nitrogen atoms on the aniline group are split to two peaks with the smaller one remaining at the same position and the larger one red shifting to 401.48 eV. This shift indicates that the state of N on the aniline groups changes through the interactions with CH₃I molecules. FT-IR spectra of CH₃I@MHP-P5Q and CH₃I@MHP display several new peaks at 982 and 1071 cm⁻¹ (Fig. 5c, d), implying the formation of new C–N bonds in these two materials. TGA results show that the weight loss below 100 °C are 7 and 10 wt% for CH₃I@MHP and CH₃I@MHP-P5Q, respectively, indicating that only a small fraction of the captured CH₃I was released at the temperature higher than its boiling point (Supplementary Fig. 29). Elemental mapping using HR-TEM energy-dispersive X-ray spectroscopy (HR-TEM-EDS) confirms that after heating CH₃I@MHP and CH₃I@MHP-P5Q at 100 °C for 2 h, iodine (CH₃I) are still found to be uniformly dispersed within the two porous polymer samples (Fig. 5e, f). According to the above results, we deduced that not only physical adsorption but also chemical adsorption of CH₃I occurred in these two adsorbents and the chemical adsorption site might be the aniline nitrogen atoms on the hydrophenazine rings.

To confirm the possible chemical adsorption process and probe the chemical adsorption site, CH₃I-appended MC (MC-CH₃I) was firstly obtained in a solution-based reaction. ¹H NMR spectrum of MC-CH₃I clearly shows the proton signals of the methyl group,

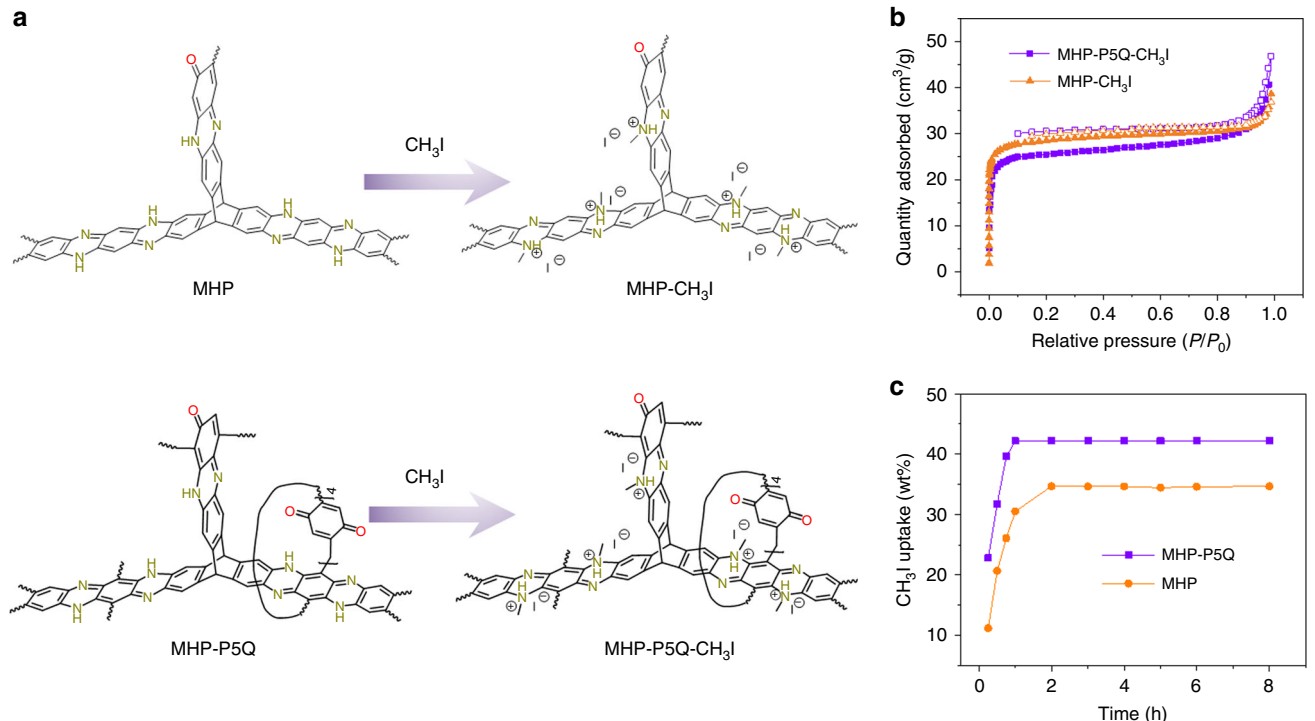

**Fig. 6 Synthesis and characterizations of CH₃I-modified MHPs. a** Synthesis of CH₃I-modified MHPs: MHP-CH₃I and MHP-P5Q-CH₃I. **b** N₂ adsorption and desorption isotherms of MHP-CH₃I (orange, $S_{BET} = 110$ m²/g) and MHP-P5Q-CH₃I (purple, $S_{BET} = 98$ m²/g) at 77 K. **c** Time-dependent CH₃I adsorption isotherm of MHP-CH₃I and MHP-P5Q-CH₃I.

confirming the strong attachment of CH₃I onto MC (Supplementary Figs. 30 and 31). To further illustrate the chemical adsorption site, CH₃I-appended MHP-P5Q (MHP-P5Q-CH₃I) and MHP (MHP-CH₃I) were synthesized through solution-based post-synthetic modifications of MHP-P5Q and MHP, respectively (Fig. 6a). Compared with MHP-P5Q and MHP, the ¹³C CP-MAS NMR spectra of the resulting MHP-P5Q-CH₃I and MHP-CH₃I show two new sharp peaks at around 36 ppm, which are corresponding to the carbons on the methyl group (Supplementary Figs. 32 and 33). Elemental analysis reveals an iodine content of 34.13 and 30.15 wt% for MHP-P5Q-CH₃I and MHP-CH₃I, respectively. These results confirm the successful graft of CH₃I onto MHP-P5Q and MHP. Meanwhile, the N 1s XPS spectrum of MHP-P5Q-CH₃I show two peaks at 399.08 and 401.28 eV, with the former corresponding to imine nitrogen atoms and the latter referring to the aniline nitrogen atoms after attachment of CH₃I (Fig. 5a). The FT-IR spectra of MHP-P5Q-CH₃I and CH₃I@MHP-P5Q are quite similar but different from the pristine MHP-P5Q (Fig. 5c). According to the above results, we can conclude that a large number of aniline groups on MHP-P5Q took part in the chemical adsorption of CH₃I while only a small amount of aniline groups remained unreactive. In contrast, the N 1s XPS spectra of both MHP-CH₃I and CH₃I@MHP show three peaks at 399.08, 400.18, and 401.28 eV, indicating that the reaction of CH₃I with MHP in solution was incomplete and the chemical adsorption of CH₃I occurred on the aniline group of MHP (Fig. 5b). The similarity in the FT-IR spectra of MHP-CH₃I and CH₃I@MHP further confirmed the above result (Fig. 5d).

Gas adsorption experiments of MHP-P5Q-CH₃I and MHP-CH₃I were further carried out to demonstrate their porosity. N₂ sorption experiments revealed Type I sorption isotherms for MHP-P5Q-CH₃I and MHP-CH₃I with their BET surface areas of 98 and 110 m²/g, respectively, which are significantly lower than the pristine MHP-P5Q and MHP (Fig. 6b). Meanwhile, the

three-step N₂ adsorption isotherm cannot be observed for MHP-P5Q-CH₃I. The results suggest that the attachment of CH₃I has a huge effect on the pore structure of MHP-P5Q and MHP, resulting in a decrease of their porosity. Moreover, the CH₃I capture capacity of MHP-P5Q-CH₃I and MHP-CH₃I were demonstrated to have a huge decrease. The maximum capacity for MHP-P5Q-CH₃I is 42.5 wt% whereas that of MHP-CH₃I is 35.2 wt% (Fig. 6c). It is thus implied that the chemical adsorption process played a vital role in the CH₃I capture.

In addition to the chemical adsorption, the physical adsorption of CH₃I in MHP-P5Q and MHP is also significant. To demonstrate the potential physical-binding site for MHP-P5Q and MHP, quantum chemical calculations were carried out using the MP2 method. Two molecules, 1,2-dihydrophenazine with –C=N– bond and 5,10-dihydrophenazine without the double bond, were chosen as MCs. The calculation results show that CH₃I forms the weak halogen bond with –N=C– in 1,2-dihydrophenazine with $d_{[N\cdots I]} = 3.39$ Å while such a halogen bond does not occur in 5,10-dihydrophenazine, whose binding energy with CH₃I is even smaller (Supplementary Fig. 34). Based on these results, one of the potential physical-binding sites for CH₃I in MHP-P5Q and MHP may be the imine group on the hydrophenazine rings through halogen bonding (–I···N=C–).

Given the fact that the amine and imine groups as chemical adsorption sites and physical-binding sites exist in both MHP-P5Q and MHP, we deduced that the enhanced CH₃I adsorption capacity in MHP-P5Q might also benefit from the pillar[5]arene skeletons embedded in the framework, which offer additional binding sites for CH₃I. To better understand the affinity between CH₃I and pillar[5]arene backbone, EtP5 without other functional groups was also chosen as a model (Supplementary Fig. 35). Single-crystal X-ray diffraction structure of CH₃I-loaded EtP5 (CH₃I@EtP5) shows that CH₃I forms a 1:2 host–guest inclusion complex with EtP5 with their methyl groups in the cavity center

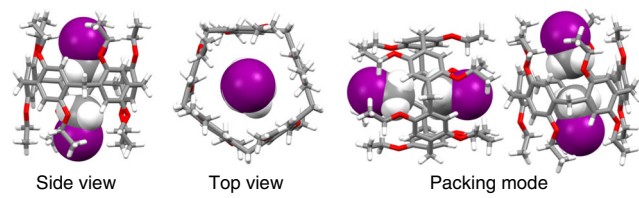

**Fig. 7 Single-crystal X-ray structure analysis.** Single-crystal X-ray structure of CH₃I-loaded EtP5 (CH₃I@EtP5).

while no $CH_3I$ molecules can be found in the extrinsic space of EtP5 (Fig. 7). The hydrogen atoms on the methyl groups are found to have CH⋯π interactions with corresponding nearest arenes on EtP5. Hence, multiple CH⋯π interactions are the main driving force for the formation of the inclusion complex. Moreover, the PXRD pattern of EtP5 after adsorption of $CH_3I$ matches well with that simulated from CH₃I@EtP5, indicating that EtP5 transform to CH₃I@EtP5 upon capture of $CH_3I$ (Supplementary Fig. 34). Thus, the limited $CH_3I$ adsorption amount in EtP5 can be attributed to the limited accessible intrinsic cavity for $CH_3I$, whereas the extrinsic space cannot be utilized for $CH_3I$ loading. However, these multiple CH⋯π interactions alone are not strong enough for the long-term storage of $CH_3I$ in EtP5. Based on above, we can conclude that compared with other MHPs, the presence of pillar[5]arene cavity in MHP-P5Q offers additional binding sites for $CH_3I$ through host–guest interactions (multiple CH⋯π interactions), thus enhancing the adsorption capacity. In contrast to EtP5, the enhanced adsorption capacity of $CH_3I$ in MHP-P5Q benefits from the abundant nitric groups in the multi-microporous framework for offering the halogen bond (–I⋯N=C–) as well as chemical adsorption sites. The synergistic effect of the multiple supramolecular interactions and the formation of chemical bonds in the unique multi-microporous structure contribute to the superior performance of MHP-P5Q in $CH_3I$ capture and storage.

In conclusion, we reported a solvent and catalyst-free mechanochemical synthesis of P5Q-derived hydrophenazine-linked multi-microporous organic polymer MHP-P5Q regardless of the poor solubility of P5Q. Benefitting from our rational design and the unique synthesis, MHP-P5Q displays a unique three-step $N_2$ sorption isotherm with three distinct pore size distribution. Compared with MHP, MHP-Cl, and MHP-Br, MHP-P5Q with similar BET surface area was demonstrated to have a superior performance in the capture and storage of $CH_3I$, a typical radioactive organic iodide waste in the nuclear industry. Mechanistic studies confirm that the rapid and high-capacity adsorption and stable storage of $CH_3I$ in MHP-P5Q come from the rigid pillar[5]arene cavity as additional binding sites for $CH_3I$ through CH⋯π interactions as well as halogen bond (–I⋯N=C–) between $CH_3I$ and imine group and chemical adsorption in the multi-microporous framework. Hence, MHP-P5Q as an adsorbent can interact with both the polar part (⋯I) and nonpolar part (⋯CH₃) of $CH_3I$ through a synergistic effect to achieve the maximization in $CH_3I$ capture and storage, which is superior to other adsorbents that only have single interactions with either the polar or the nonpolar part of $CH_3I$. A future challenge may be targeted to the incorporation of other supramolecular macrocycles with the reservation of their cavities into robust porous frameworks for task-specific applications.

## Methods

**Synthesis of MHP.** Benzoquinone (0.108 g; 1 mmol) and triptycenehexaamine hexahydrochloride (0.374 g; 0.67 mmol) were added into a 15 mL stainless steel grinding jar (33 mm diameter) with three steel balls. The mixture was then ground

for 30 min in a Retsch MM400 grinder mill operating at 30 Hz. The solids were washed with saturated NaHCO₃ solution for two times and methanol for four times, which were dried at 100 °C for 12 h to give dark brown powders (0.148 g).

**Synthesis of MHP-Cl.** The synthesis of MHP-Cl is pretty similar to that of MHP. In particular, 2,5-dichloro-1,4-benzoquinone (0.176 g; 1 mmol) and triptycene-hexaamine hexahydrochloride (0.374 g; 0.67 mmol) were added into a 15-mL stainless steel grinding jar (33 mm diameter) with three steel balls. The mixture was then ground for 30 min in a Retsch MM400 grinder mill operating at 30 Hz. The solids were washed with saturated NaHCO₃ solution for two times and methanol for four times, which were dried at 100 °C for 12 h to give dark brown powders (0.218 g).

**Synthesis of MHP-Br.** The synthesis of MHP-Br is pretty similar to that of MHP. In particular, 2,5-dibromo-1,4-benzoquinone (0.265 g; 1 mmol) and triptycene-hexaamine hexahydrochloride (0.374 g; 0.67 mmol) were added into a 15-mL stainless steel grinding jar (33 mm diameter) with three steel balls. The mixture was then ground for 30 min in a Retsch MM400 grinder mill operating at 30 Hz. The solids were washed with saturated NaHCO₃ solution for two times and methanol for four times, which were dried at 100 °C for 12 h to give dark brown powders (0.276 g).

**Synthesis of MHP-P5Q.** The synthesis of MHP-P5Q is pretty similar to that of MHP. In particular, P5Q (0.150 g; 0.25 mmol) and triptycenehexaamine hexahydrochloride (0.468 g; 0.838 mmol) were added into a 15-mL stainless steel grinding jar (33 mm diameter) with three steel balls. The mixture was then ground for 30 min in a Retsch MM400 grinder mill operating at 30 Hz. The solids were washed with saturated NaHCO₃ solution for two times and methanol for four times, which were dried at 100 °C for 12 h to give dark brown powders (0.204 g).

## Data availability

The X-ray crystallographic coordinates for structures reported in this study have been deposited at the Cambridge Crystallographic Data Centre (CCDC), under deposition number 1948606. These data can be obtained free of charge from The Cambridge Crystallographic Data Centre via www.ccdc.cam.ac.uk/data_request/cif. All other data supporting the findings of this study are available from the article and its Supplementary Information files or available from the corresponding authors upon reasonable request.

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

## Acknowledgements

This work was supported by the U.S. Department of Energy, Office of Science, Office of Basic Energy Sciences, Chemical Sciences, Geosciences, and Biosciences Division.

## Author contributions

K.J. and S.D. conceived the project and designed the experiments. K.J., Y.Z., and R.Z. conducted the synthesis and $CH_3I$ capture experiments. K.J., W.G., Z.Y., and H.C. performed gas sorption, XRD and XPS experiments and analyzed the data. B.L. and D.J. performed the theoretical calculation and analyzed data. K.J., Q.S., and S.D. co-wrote the paper. K.J., B.L., D.J., Q.S., F.H., and S.D. discussed the results and commented on the manuscript.

## Competing interests

The authors declare no competing interests.
