## [Peer Review File · Nature Communications]

Reviewers' comments:

Reviewer #1 (Remarks to the Author):

Thank you for the invitation to review the manuscript "Mechanochemical Synthesis of Pillar[5]quinone Derived 1 Multi-Microporous Organic Polymers for Radioactive Organic Iodide Capture and Storage", co-authored by Jie et al.

In this manuscript, the authors present an innovative and apparently highly successful synthesis of new microporous organic polymers based on hydrophenazine units (MHPs) using mechanochemistry, and use them for binding of CH₃I, a substance relevant in storage and transport of radioactive iodine. While I congratulate the authors on the mechanochemical work, and interesting materials design, I am not sure that the methyl iodide binding aspect is particularly novel, especially as it appears not to be based on a particular structural aspect of the MHP materials, but mostly on chemisorption. Also, while the synthetic and materials characterization aspect of the work is interesting and novel, I find that there are a number of issues that still must be addressed. Consequently, I do not think this work is ready for publication. I am highlighting below some of the problems and would recommend the authors to perform a major overhaul of the text, as well as re-consider the interpretation of their data.

1) The authors observe binding in the product MHP materials both by chemical sorption – formation of methylammonium cation/iodide anion pairs, as well as physical sorption of methyl iodide. I find the formation of methylammonium salts by exposure of NH groups to CH₃I to be quite usual, and am wondering if equally well chemisorption of CH₃I would be achieved using a simpler, small molecule system, such as phenazine?

2) I am also suspicious of the authors interpretation of desorption data – if chemisorption of CH₃I happens by formation of R₂NHCH₃⁺/I⁻ salt systems, would it not be possible that heating of such materials would lead to loss of HI, according to the equation:

Would the authors be able to provide a mass spectrometry analysis of the gas released during desorption?

3) The chemical reaction in Scheme 1a must be balanced, as there obviously must be byproducts of the condensation reaction, but also of the oxidation of the aromatic ring. In that respect, the authors should explain how did the C-H group of the disubstituted p-quinone get replaced? The authors note that such a reaction was never previously observed using mechanochemistry – so more reason to explain what happened? My guess is that the reaction contains a sizeable excess of benzoquinone, which facilitates the oxidative C-H + H-N coupling to give a C-N link and H₂O/catechol byproducts...? Or is H₂ a reaction byproduct? Or is there participation of oxygen from air?

4) Similarly to the above comment, the reaction equation in Supplementary Section 3.2 is incorrect. As the authors note the use of the quinone and phenylenediamine in a 1:2 ratio, there must be hydrogen being evolved, or participation of oxygen? In case of the latter, how did the authors ensure/control there was sufficient oxygen in the reaction mixture?

5) Line 97: It is not clear what do the authors describe as "2-fold benzoquinone"? That is not conventional nomenclature?

6) Line 105: the sentence "solid-state ¹³C cross-polarization magic-angle spinning nuclear magnetic resonance (¹³C CP-MAS NMR) at the molecular level" is awkward – why is it necessary

to stress that this is on a "molecular level"? With respect to that, would it not be useful to perform NMR experiments also on 15N? That should provide more info, as there are fewer nitrogen atoms per polymer subunit. It would also nicely complement the XPS data.

7) There are some residual items from manuscript preparation. For example, the caption to supplementary figure S9 is: "PXRD pattern of MHP. (degrees should be degree)"? This makes me think that this submission is not quite fully polished yet.

8) It would be useful if figure captions in the supplementary information would be more informative. For example, figures S10, S13, S16, S21, S27 all show TG curves, but with no interpretation or numerical values for the size and temperature of observed mass loss steps?

9) How did the authors deduce the signal assignments for ¹³C solid-state NMR spectra?

10) Line 253: the language describing the process of CH₃I sorption by methylation of nitrogen atoms is incorrect: "This shift indicates that the valence of N on the aniline groups is increased through the interactions with CH₃I molecules." The concept of nitrogen changing "valence" in forming ammonium cations is archaic and should not be used.

11) The compound Et5P is prominent in this study, but the authors did not provide a structure. It is chemically very different from their MHPs, so I am also not sure it is very relevant as a material for comparison of CH₃I sorption properties.

12) Line 321: This sentence is confusing: "Given the fact that the two different nitric groups as chemical adsorption sites and physical binding sites exist in both MHP-P5Q and...". What do the authors mean by "nitric group"? As far as I can see, there are only amine and imine sites?

Reviewer #2 (Remarks to the Author):

The authors describe in this paper for the first time a solvent and catalyst-free mechanochemical synthesis of pillar[5]quinone derived multimicroporous organic polymers with hydrophenazine linkages, which show a unique 3-step N₂ adsorption isotherm and good performance in radioactive iodomethane capture and storage. The paper is clearly and interestingly written, and in my opinion it is certainly worth publishing in Nature Communications. Nevertheless, I have a number of minor comment, which would be nice to address prior to acceptance.

1- To increase the value of this paper, the authors should compare the adsorption performance of their materials in radioactive iodomethane capture to those of activated porous carbon impregnated with triethylenediamine and KI and zeolites exchanged with silver that are used actually in nuclear industry. (See ref J. Huve, et al., Porous sorbents for the capture of iodine radioactive compounds: A review, RSC Advances, 2018, 8, 29248.)

2- Others compounds can also be present during a severe nuclear accident, such as Volatile Organic Compounds (VOCs), CO_x (CO₂ and CO), SO_x (SO₂ and SO₃), water vapor. Do the authors have an idea if these compounds could affect significantly the iodine adsorption performances of their porous materials?

3- During nuclear accident, the temperature can increase. According to the authors their materials are stable up to 350 °C but do they know if the adsorption capacities of their materials for iodomethane decrease for temperature above 100°C.

4- Do the authors know if MHP-P5Q materials and the other materials synthesized in this paper could be affected by γ radiation??

Reviewer #3 (Remarks to the Author):

This work was contributed by Dai et al. describes an interesting research finding of Multi-Microporous Organic Polymer which is synthesized without solvent and catalyst. In this work, MHP-P5Q constructed by pillar[5]quinone (P5Q) and hydrophenazine linkages under mechanochemical synthesis shows great adsorption and storage of CH₃I, which is considered as a radioactive organic iodide. The Mechanistic studies explains that the rigid pillar[5]arene cavity in MHP-P5Q endowing extra binding sites and the halogen bond to CH₃I, combining with the chemical adsorption in the multi-microporous MHP-P5Q may be the key point why MHP-P5Q stands out of analogous microporous organic polymers which are constructed by simple 2-fold benzoquinones instead and hydrophenazine linkages. Detailed Characterizations of materials have been carried out and credible explanation has been discussed carefully. So, based on the interesting finding and excellent results, I think the work will raise interest in porous organic polymers and is suitable for publication in Nature Communication after addressing the following minor issues.

1. The recycle usage of absorbent is another key factor under consideration, was any loss of adsorption capacity of CH₃I observed after first-time adsorption and desorption?
2. The real radioactive CH₃I adsorption will be underwent in severe circumstances, so could the author provide more stability testing of MHP-P5Q in order to verify the practical utilization of it in the real industry?
3. In the line of 87 and 88, the names of polymers are MOP-Cl, MOP-Br and MOP-P5Q, respectively, which is confusing. Please correct them if they are not the final names.
4. All the figures are not clear in the main text, which should be replaced with higher resolution ones.
5. Some closely related review article regarding POP for radionuclide sequestration is suggested to be cited: Trends in Chemistry 2019, 1, 292-303.

Comments Reviewer 1:

1. *In this manuscript, the authors present an innovative and apparently highly successful synthesis of new microporous organic polymers based on hydrophenazine units (MHPs) using mechanochemistry, and use them for binding of CH₃I, a substance relevant in storage and transport of radioactive iodine. While I congratulate the authors on the mechanochemical work, and interesting materials design, I am not sure that the methyl iodide binding aspect is particularly novel, especially as it appears not to be based on a particular structural aspect of the MHP materials, but mostly on chemisorption. Also, while the synthetic and materials characterization aspect of the work is interesting and novel, I find that there are a number of issues that still must be addressed. Consequently, I do not think this work is ready for publication. I am highlighting below some of the problems and would recommend the authors to perform a major overhaul of the text, as well as re-consider the interpretation of their data.*

We sincerely thank reviewer 1 for the positive comments on the synthesis and materials characterization aspect of the work. For the significant concerns regarding the synthesis, we have made corresponding revisions and provided responses below to improve the manuscript. For the concerning about CH₃I adsorption part, while chemisorption of CH₃I did occur in all four MHPs, physical adsorption also played a vital role in the capture and storage performance and we have demonstrated the superiority of P5Q-MHP containing pillararene skeletons. Specifically, P5Q-MHP can interact with both the polar part ($\cdots\text{I}$, chemisorption and halogen bond) and nonpolar part ($\cdots\text{CH}_3$, $\text{CH}\cdots\pi$ interactions) of CH₃I through a synergistic effect to achieve the maximization in CH₃I capture and storage, which is superior to other MHPs that only have interactions with the polar part ($\cdots\text{I}$) of CH₃I.

2. *The authors observe binding in the product MHP materials both by chemical sorption – formation of methylammonium cation/iodide anion pairs, as well as physical sorption of methyl iodide. I find the formation of methylammonium salts by exposure of NH groups to CH₃I to be quite usual, and am wondering if equally well chemisorption of CH₃I would be achieved using a simpler, small molecule system, such as phenazine?*

Many thanks for the concern. Here, we used model compound MC to demonstrate whether a small fragment of the material could achieve chemisorption. However, the ¹H NMR spectra of MC before and after adsorption of CH₃I vapor are almost the same (Figure 1), indicating that chemisorption did not occur. We inferred that the reason why the chemisorption of CH₃I occurred in the MHPs benefitted from both the NH groups and the porosity of MHPs. Since MHPs are porous, CH₃I can easily enter into the framework of MHPs to react with NH groups. However, the loss of accessible pores in nonporous crystals of MC might drive CH₃I away from NH groups inside these crystals. In this regard, CH₃I can react with MC in the solution phase but

not in the solid-vapor phase.

Figure 1. ^1H NMR spectra (400 MHz, $\text{DMSO-}d_6$, 293 K) of MC before (top) and after (below) adsorption of CH_3I vapor.

3. *I am also suspicious of the authors interpretation of desorption data – if chemisorption of CH_3I happens by formation of $\text{R}_2\text{NHCH}_3^+/\text{I}^-$ salt systems, would it not be possible that heating of such materials would lead to loss of HI, according to the equation:*

Would the authors be able to provide a mass spectrometry analysis of the gas released during desorption?

Many thanks for the valuable comment. We here provide two mass spectra of both pure CH_3I and the gas released from the materials (heating at $100\text{ }^\circ\text{C}$), which are shown below. The results show that the only species that can be detected is I^- ($\text{M}/\text{S}^- = 126.9$). In the regard, we cannot tell whether the released gas from our material is HI or CH_3I from mass spectrometry analysis. However, we can conclude from our EDS mapping and TGA results that the release gas at $100\text{ }^\circ\text{C}$ should be the physically adsorbed CH_3I . If heating at $100\text{ }^\circ\text{C}$ could produce HI, no more I species could be detected by EDS mapping. Even if HI could be produced upon heating, the temperature should be much higher than $100\text{ }^\circ\text{C}$. This, on the other hand, does not affect the conclusions in the work as we are more focused on the CH_3I capture and storage performance at room temperature or below $100\text{ }^\circ\text{C}$.

Figure 2. Mass spectra of pure CH_3I (top) and the gas released from the material (heating at 100°C , bottom).

4. *The chemical reaction in Scheme 1a must be balanced, as there obviously must be byproducts of the condensation reaction, but also of the oxidation of the aromatic ring. In that respect, the authors should explain how did the C-H group of the disubstituted p-quinone get replaced? The authors note that such a reaction was never previously observed using mechanochemistry – so more reason to explain what happened? My guess is that the reaction contains a sizeable excess of benzoquinone, which facilitates the oxidative C-H + H-N coupling to give a C-N link and H_2O /catechol byproducts...? Or is H_2 a reaction byproduct? Or is there participation of oxygen from air?*

Many thanks for the valuable question. In fact, the original reaction between BQ and diamine contains two parts: (1) condensation reaction between amine and $\text{O}=\text{C}$; (2) Michael addition reaction between amine and $-\text{H}-\text{C}=\text{C}-\text{H}-$. As for the Michael addition, the formed C-C single bond can be easily oxidized to C=C bond again in the presence of O_2 in the air. So the only byproduct in the synthesis is H_2O for both the condensation and Michael addition processes. We also confirmed that in a solution-based reaction, model compound (MC) cannot be obtained in the absence of air but could be made upon exposure to air. As suggested, we have modified scheme 1a in the manuscript. For the details of such an aza-ring formation reaction, it can be found in the following literature: *Synlett* **25**, 495 (2014). A proposed mechanism based on the synthesis of model compound (MC) is also given below:

5. *Similarly to the above comment, the reaction equation in Supplementary Section 3.2 is incorrect. As the authors note the use of the quinone and phenylenediamine in a 1:2 ratio, there must be hydrogen being evolved, or participation of oxygen? In case of the latter, how did the authors ensure/control there was sufficient oxygen in the reaction mixture?*

Many thanks for the valuable question. Accordingly, we have modified the figures in Supplementary Section 3.2. As stated above, oxygen in the air took part in the reaction, which resulted in the byproduct of water. In the reaction, we only controlled the quantity of the starting materials, which was quite small compared with the remaining space of the milling jar. Thus, there was enough oxygen in the jar participating in the reaction. Meanwhile, the milling jar was not completely sealed, which allowed for the air to squeeze into the jar. In this regard, the reaction processed in the jar was successful.

6. *Line 97: It is not clear what do the authors describe as "2-fold benzoquinone"? That is not conventional nomenclature?*

Many thanks for the question. In the manuscript, triptycenhexamine is described as a "3-fold crosslinker" because it has 3 reactive sites for the synthesis of MHPs. In contrast, benzoquinone has 2 reactive sites and pillar[5]quinone has 10 reactive sites for the synthesis of MHPs. So here we describe simple benzoquinones as "2-fold benzoquinones" as a comparison to the "10-fold pillar[5]quinone". In order to make it clear, we change the description "2-fold benzoquinones" to "benzoquinones (2-fold monomers)".

7. *Line 105: the sentence "solid-state ^{13}C cross-polarization magic-angle spinning nuclear magnetic resonance (^{13}C CP-MAS NMR) at the molecular level" is awkward – why is it necessary to stress that this is on a "molecular level"? With respect to that, would it not be useful to perform NMR experiments also on ^{15}N ? That should provide more info, as there are fewer nitrogen atoms per polymer subunit. It would also nicely complement the XPS data.*

Many thanks for the valuable suggestions. We have deleted the statement “at the molecular level” in the revised manuscript. In fact, we had performed ^{15}N NMR experiments several times before the submission of the manuscript even if we had been told by the NMR technician that ^{15}N NMR spectra were extremely difficult to acquire. The spectra were indeed not clear even after running for several days. As suggested by the reviewer, we again tried our best to perform ^{15}N NMR experiments for a longer time to obtain a clear spectrum, but still failed at last. Despite of this, we believe that elemental analysis as well as XPS could well interpret the N state in the polymer.

8. *There are some residual items from manuscript preparation. For example, the caption to supplementary figure S9 is: "PXRD pattern of MHP. (degrees should be degree)"? This makes me think that this submission is not quite fully polished yet.*

Many thanks for the valuable comment. We do apologize about the mistake that we made in the manuscript. This has now been addressed.

9. *It would be useful if figure captions in the supplementary information would be more informative. For example, figures S10, S13, S16, S21, S27 all show TG curves, but with no interpretation or numerical values for the size and temperature of observed mass loss steps?*

Many thanks for the valuable suggestions. We have now added more information into the figures and captions of Supplementary Figure S10, S13, S16, S21, S27. The details can be now found in the revised Supplementary Information.

10. *How did the authors deduce the signal assignments for ^{13}C solid-state NMR spectra?*

Many thanks. We deduced the signal assignments of the ^{13}C solid-state NMR spectra according to several literature as well as with the aid of ChemDraw software. Specifically, some typical ^{13}C signals such as carbonyl groups could be assigned easily; others were assigned by drawing a fragment of the polymer in ChemDraw and using the function “predicting ^{13}C -NMR shifts”.

11. *Line 253: the language describing the process of CH3I sorption by methylation of nitrogen atoms is incorrect: "This shift indicates that the valence of N on the aniline groups is increased through*

the interactions with CH₃I molecules." The concept of nitrogen changing "valence" in forming ammonium cations is archaic and should not be used.

Many thanks for the valuable suggestion. We have now modified the sentence as follows: "This shift indicates that the state of N on the aniline groups changes through the interactions with CH₃I molecules."

12. *The compound Et5P is prominent in this study, but the authors did not provide a structure. It is chemically very different from their MHPs, so I am also not sure it is very relevant as a material for comparison of CH₃I sorption properties.*

Many thanks for the valuable comments. We here provide the chemical and crystal structures of EtP5 in the Supplementary Section 6.5. The reason why we chose EtP5 as a contrast is as follows: 1) As we describe in the introduction part, pillararene crystals can be used as adsorption materials, which, however, have several disadvantages such as low adsorption capacity and rate. We want to demonstrate that by embedding pillararene skeletons into porous frameworks, these disadvantages can be overcome. Since EtP5 and P5Q has the same cavity size, we chose EtP5 crystals as a contrast to P5Q-MHP; 2) The enhanced CH₃I adsorption capacity in MHP-P5Q over other MHPs may also benefit from the pillar[5]arene skeletons embedded in the framework, which offer additional binding sites for CH₃I. So EtP5 without other functional groups was also chosen as a model to better understand the affinity between CH₃I and pillar[5]arene backbone.

13. *Line 321: This sentence is confusing: "Given the fact that the two different nitric groups as chemical adsorption sites and physical binding sites exist in both MHP-P5Q and...". What do the authors mean by "nitric group"? As far as I can see, there are only amine and imine sites?*

Many thanks for the valuable question. In fact, here two different nitric groups refer to amine and imine sites as the reviewer mentioned. To make it clearer, the sentence was changed to "Given the fact that the amine and imine groups as chemical adsorption sites and physical binding sites exist in both MHP-P5Q and..."

Comments Reviewer 2:

1. *The authors describe in this paper for the first time a solvent and catalyst-free mechanochemical synthesis of pillar[5]quinone derived multimicroporous organic polymers with hydrophenazine linkages, which show a unique 3-step N₂ adsorption isotherm and good performance in radioactive iodomethane capture and storage. The paper is clearly and interestingly written, and in my opinion it is certainly worth publishing in Nature Communications. Nevertheless, I have a number of minor comment, which would be nice to address prior to acceptance.*

We thank reviewer 2 for the positive comments on our work.

2. *To increase the value of this paper, the authors should compare the adsorption performance of their materials in radioactive iodomethane capture to those of activated porous carbon impregnated with triethylenediamine and KI and zeolites exchanged with silver that are used actually in nuclear industry. (See ref J. Huve, et al., Porous sorbents for the capture of iodine radioactive compounds: A review, RSC Advances, 2018, 8, 29248.)*

Many thanks. As suggested, we tested the CH₃I adsorption performance of activated porous carbon impregnated with triethylenediamine (TED@AC) and silver functionalized zeolites (including ZSM-5, 13X, and mordenite, which are named as Ag⁺@ZSM-5, Ag⁺@13X, Ag⁺@MOR, and Ag⁰@MOR for short, respectively). The CH₃I uptake amounts at room temperature in TED@AC, Ag⁺@ZSM-5, Ag⁺@13X, Ag⁺@MOR, and Ag⁰@MOR are 50.6, 27.3, 47.4, 30.7, and 24.3 wt%, respectively. In comparison, MHP-P5Q shows a remarkable higher uptake amount than other MHPs and these benchmark materials (Figure 3 or Supplementary Figure 27).

Figure 3. Comparing the saturated CH₃I uptake in MHPs and selected benchmark sorbent materials at room temperature.

3. *Other compounds can also be present during a severe nuclear accident, such as Volatile Organic Compounds (VOCs), CO_x (CO₂ and CO), SO_x (SO₂ and SO₃), water vapor. Do the authors have an idea if these compounds could affect significantly the iodine adsorption performances of their porous materials? .*

Many thanks for the concern. Acidic gases such as CO₂, SO₂ or SO₃ will be likely to act as competitive guests with CH₃I because they have hydrogen bonding interactions with the amine groups in these materials. This may induce a decrease on CH₃I adsorption performance. Other VOCs may have little effect on CH₃I adsorption performance due to the loss of binding sites in MHPs. Compared with simple MHPs, we have demonstrated that the enhanced CH₃I uptake capacity in P5Q-MHP also benefits from pillar[5]arene cavity embedded in MHP-P5Q as additional binding sites for CH₃I through host-guest interactions (multiple CH $\cdots\pi$ interactions). However, these are not binding sites for acidic gases. Even though the presence of acidic gases may decrease the CH₃I adsorption in MHPs, their impact on CH₃I adsorption in MHP-P5Q may be the smallest, which will also demonstrate the superiority of MHP-P5Q in CH₃I adsorption.

4. *During nuclear accident, the temperature can increase. According to the authors their materials are stable up to 350 °C but do they know if the adsorption capacities of their materials for iodomethane decrease for temperature above 100 °C.*

Many thanks. As suggested, we performed CH₃I adsorption experiments using MHPs and EtP5 at 100 °C. As shown in Figure 4 or Supplementary Figure 28, all these adsorbents show a decrease in CH₃I uptake at 100 °C. This may be induced by the decrease in physical adsorption amount. Especially for EtP5, it cannot adsorb CH₃I at 100 °C.

Figure 4. Comparison of CH₃I uptake in MHPs and EtP5 at 25 °C and 100 °C.

5. *Do the authors know if MHP-P5Q materials and the other materials synthesized in this paper*

could be affected by γ radiation??

Many thanks. These materials should be stable upon γ radiation as there are no functional groups that are responsive to γ radiation. To verify this, we exposed MHPs in γ radiation for 1 h. The results show that the porosity of these materials was not changed.

Comments Reviewer 3:

- 1. This work was contributed by Dai et al. describes an interesting research finding of Multi-Microporous Organic Polymer which is synthesized without solvent and catalyst. In this work, MHP-P5Q constructed by pillar[5]quinone (P5Q) and hydrophenazine linkages under mechanochemical synthesis shows great adsorption and storage of CH₃I, which is considered as a radioactive organic iodide. The Mechanistic studies explains that the rigid pillar[5]arene cavity in MHP-P5Q endowing extra biding sites and the halogen bond to CH₃I, combining with the chemical adsorption in the multi-microporous MHP-P5Q may be the key point why MHP-P5Q stands out of analogous microporous organic polymers which are constructed by simple 2-fold benzoquinones instead and hydrophenazine linkages. Detailed Characterizations of materials have been carried out and credible explanation has been discussed carefully. So, based on the interesting finding and excellent results, I think the work will raise interest in porous organic polymers and is suitable for publication in Nature Communication after addressing the following minor issues.*

We thank reviewer 3 for the positive comments on our work.

- 2. The recycle usage of absorbent is another key factor under consideration, was any loss of adsorption capacity of CH₃I observed after first-time adsorption and desorption?*

Many thanks. In this case, both physical adsorption and chemisorption of CH₃I occurred in MHPs. For the CH₃I that is chemically adsorbed, it cannot be released in the desorption process. As a result, the performance of MHP adsorbents will decrease dramatically after the first cycle. However, in this case, we believe that the reliable storage of the radioactive CH₃I using MHPs, which partly benefits from the chemisorption, is a more important factor than the recycle usage in the real nuclear industry.

- 3. The real radioactive CH₃I adsorption will be underwent in severe circumstances, so could the author provide more stability testing of MHP-P5Q in order to verify the practical utilization of it in the real industry?*

Many thanks for the suggestion. We have tested the stability of MHP-P5Q in different harsh circumstances, which was determined by the BET surface area. As can be seen from Table 1

(Table S3), MHP-P5Q is demonstrated to be stable in boiling water, strong base environment and upon treatment with water steam. However, the BET surface area of MHP-P5Q will decrease in weak acid conditions, which probably results from the formation of ammonium salts system. The porosity of MHP-P5Q can be recovered by washing with saturated NaHCO₃ solution. MHP-P5Q will completely be destroyed in strong acid conditions (1 M HCl), which cannot be recovered. We believe that MHP-P5Q is stable enough for the practical utilization in the real industry.

Table 1. BET surface area of MHP-P5Q in different conditions.

	Boiling water	Water steam	1 M NaOH	0.1 mM HCl	1 M HCl
BET surface area (m ² /g)	302	293	295	226	7

4. *In the line of 87 and 88, the names of polymers are MOP-Cl, MOP-Br and MOP-P5Q, respectively, which is confusing. Please correct them if they are not the final names.*

Many thanks. These have been addressed in the manuscript.

5. *All the figures are not clear in the main text, which should be replaced with higher resolution ones.*

Many thanks for the suggestion. The PDF files are generated automatically upon submission of word files. In this process, the figures with high resolution in the main text might become unclear. We believe that the editors will solve the problem in the end.

6. *Some closely related review article regarding POP for radionuclide sequestration is suggested to be cited: Trends in Chemistry 2019, 1, 292-303.*

Many thanks for the suggestion. This has been cited in the revised manuscript.

REVIEWERS' COMMENTS:

Reviewer #1 (Remarks to the Author):

Thank you for the opportunity to review this work again. I believe that the authors have provided satisfactory replies to queries of all three Referees and I would support the publication of this work in Nature Communications.

However, several equations still need to be corrected... Please correct the reaction scheme on Page S12 of the Supplementary Materials, to clearly show that oxygen is also participating in the reaction. Also, what happens with chloride ions in this reaction? Same on page S14, as well as S16, S20.

Finally, just as a minor point of interest - have the authors also looked at the positive mode in their MS study?

Reviewer #2 (Remarks to the Author):

The authors have taken into account all the referee remarks. The paper can be accepted in its current form for publication.

Reviewer #3 (Remarks to the Author):

The authors have satisfactorily addressed all the comments from the reviewer and no further revision is needed.

Comments Reviewer 1:

1. *Thank you for the opportunity to review this work again. I believe that the authors have provided satisfactory replies to queries of all three Referees and I would support the publication of this work in Nature Communications.*

We sincerely thank reviewer 1 for the positive comments on our revised manuscript.

2. *However, several equations still need to be corrected... Please correct the reaction scheme on Page S12 of the Supplementary Materials, to clearly show that oxygen is also participating in the reaction. Also, what happens with chloride ions in this reaction? Same on page S14, as well as S16, S20.*

Many thanks for the valuable suggestions. We have modified the reaction schemes on page S12, S14, S16 and S20. For the chloride ions, most of them exist as hydrogen chloride while others remain as chloride ions due to the reaction of hydrogen chloride with the amine groups on the polymers. All of them will be removed upon washing with saturated NaHCO₃ solution.

3. *Finally, just as a minor point of interest - have the authors also looked at the positive mode in their MS study?*

Many thanks for the valuable comment. We have looked at the positive mode, but we could only detect the CH₃⁺ signal. That is why we used the negative mode in MS study.

Comments Reviewer 2:

1. *The authors have taken into account all the referee remarks. The paper can be accepted in its current form for publication.*

We sincerely thank reviewer 2 for the positive comments on our revised manuscript.

Comments Reviewer 3:

1. *The authors have satisfactorily addressed all the comments from the reviewer and no further revision is needed.*

We sincerely thank reviewer 3 for the positive comments on our revised manuscript.